# SavvyCNV: Genome-wide CNV calling from off-target reads

**Thomas W. Laver** [ID], **Elisa De Franco** [ID], **Matthew B. Johnson** [ID], **Kashyap A. Patel** [ID], **Sian Ellard** [ID], **Michael N. Weedon** [ID], **Sarah E. Flanagan** [ID], **Matthew N. Wakeling** [ID] *

Institute of Biomedical & Clinical Science, University of Exeter, Exeter, United Kingdom

* m.wakeling@exeter.ac.uk

**Data Availability Statement:** The SavvyCNV tool and the code used to run the benchmarking comparisons are freely available on github. The tool is available at https://github.com/rdemolgen/SavvySuite. The code used to run the

## Abstract

Identifying copy number variants (CNVs) can provide diagnoses to patients and provide important biological insights into human health and disease. Current exome and targeted sequencing approaches cannot detect clinically and biologically-relevant CNVs outside their target area. We present SavvyCNV, a tool which uses off-target read data from exome and targeted sequencing data to call germline CNVs genome-wide. Up to 70% of sequencing reads from exome and targeted sequencing fall outside the targeted regions. We have developed a new tool, SavvyCNV, to exploit this 'free data' to call CNVs across the genome. We benchmarked SavvyCNV against five state-of-the-art CNV callers using truth sets generated from genome sequencing data and Multiplex Ligation-dependent Probe Amplification assays. SavvyCNV called CNVs with high precision and recall, outperforming the five other tools at calling CNVs genome-wide, using off-target or on-target reads from targeted panel and exome sequencing. We then applied SavvyCNV to clinical samples sequenced using a targeted panel and were able to call previously undetected clinically-relevant CNVs, highlighting the utility of this tool within the diagnostic setting. SavvyCNV outperforms existing tools for calling CNVs from off-target reads. It can call CNVs genome-wide from targeted panel and exome data, increasing the utility and diagnostic yield of these tests. SavvyCNV is freely available at https://github.com/rdemolgen/SavvySuite.

## Author summary

We have created SavvyCNV, a new tool for calling genetic variants. Large regions of the genome can be deleted or duplicated–these variants can have important consequences, for example causing a patient's genetic disease. However, many standard genetic tests only target a small fraction of the genome and will miss variants outside of these regions. Therefore, we developed a tool to exploit sequencing data which falls outside of these regions (due to flaws in the targeting process) to call large deletions and duplications. This allows large deletions and duplications to be detected anywhere in the genome. Researchers and diagnostic laboratories can use this tool to discover more genetic variants by re-analysing their sequencing data.

benchmarking comparisons is available at: https://github.com/exeter-matthew-wakeling/SavvyCNV_benchmarking. Our study uses the ICR96 data set for benchmarking, which is publicly available and can be accessed through the European-Genome phenome Archive (EGA) under the accession number EGAS00001002428. The dataset of 2591 samples referred to the molecular genetics department at the Royal Devon and Exeter Hospital for genetic testing cannot be shared due to patient confidentiality issues, as the genotype data could be used to identify individuals and so cannot be made openly available. Requests for access to the anonymised data by researchers will be considered following an application to the Genetic Beta Cell Research Bank (https://www.diabetesgenes.org/current-research/genetic-beta-cell-research-bank/) with proposals reviewed by the Genetic Data Access Committee.

**Funding:** This project utilised high-performance computing funded by the UK Medical Research Council (MRC) Clinical Research Infrastructure Initiative (award number MR/M008924/1). TWL is the recipient of a Lectureship, and MBJ and MNW an Independent Fellowship from the Exeter Diabetes Centre of Excellence funded by Research England's Expanding Excellence in England (E3) fund. EDF is a Diabetes UK RD Lawrence Fellow. The funders had no role in study design, data collection and analysis, decision to publish, or preparation of the manuscript.

**Competing interests:** The authors have declared that no competing interests exist.

This is a *PLOS Computational Biology* Software paper.

## Introduction

Copy number variants (CNVs) are an important class of genetic variant. They can cause monogenic disease [1,2], are associated with polygenic traits [3] and may exert pharmacogenetic effects [4]. CNVs are structural rearrangements where bases are gained (duplication) or lost (deletion) from the genome causing an altered copy number compared to the reference.

The importance of CNVs is highlighted by the role they play in many diseases, including cancers [5], autism [6], developmental disorders [7], and heart disease [8]. Single or partial gene deletions can cause disease where haploinsufficiency would result in the disease phenotype. For example, both single nucleotide variants and whole gene deletions of *PKD1* cause polycystic kidney disease [1]. Duplications can also cause disease as a result of gene disruption at the site of insertion or through increased gene expression. For example, paternal duplication of the chromosome 6q24 region causes neonatal diabetes by overexpression of the imprinted gene *PLAGL1* [2,9]. Larger CNVs are likely to cause syndromic disease as they affect multiple genes. An extreme case is Down syndrome where duplication of chromosome 21 results in characteristic facial features and intellectual disability [10].

CNVs can be detected by a range of methods. In the clinical setting DNA microarrays are routinely used to detect larger rearrangements whilst multiplex ligation-dependent probe amplification (MLPA) is often used to detect single or partial gene CNVs [11]. With next generation sequencing (NGS) increasingly employed to investigate genetic variation, the detection of CNVs from NGS data has become increasingly important. While genome sequencing is the optimal method to capture all sequence variation across the genome, due to speed and cost exome sequencing and targeted NGS panels are still the most commonly used testing methods, particularly as a first line test in clinical diagnostic laboratories.

Many methods have been published to call CNVs from exome and targeted gene panel data [12]. These are designed to detect CNVs within the genes which are targeted by the assay, however biologically interesting and disease causing CNVs will often fall outside of the targeted regions. Even where existing methods are able to identify that a particular gene is deleted/duplicated, they will not necessarily be able to map the extent of the CNV, as the breakpoints will often be located outside the targeted regions.

Current approaches to gene targeting for NGS are imperfect. Samuels *et al* reported that between 40% and 60% of sequence reads generated map outside the target regions [13]. This in effect produces ultra-low depth whole genome sequence data. While there is insufficient information (<1X coverage) to call single nucleotide variants over the untargeted region, this 'off-target' data can be exploited to call large CNVs. The very low average read depth across the off-target regions is also why split reads (where reads map to either side of a breakpoint) cannot be used to detect CNVs. Instead they must be detected using read depth changes over a wide area. The ability to use off-target reads to call CNVs across the genome increases the diagnostic utility of targeted next-generation sequencing panels and also allows for more accurate mapping of CNVs where breakpoints fall outside of the targeted regions. Previous tools have been designed to detect CNVs in off-target reads from exome data [14] and large targeted panels [15,16]. As such sequencing panels target a larger region of the genome they require a large amount of sequencing and thus produce a relatively large number of off-target reads. However, no tool has been designed to call CNVs using off-target reads from small targeted panels (<100 genes targeted).

We have developed a new tool, SavvyCNV, for calling germline CNVs from off-target reads. It is able to call off-target CNVs from small targeted panels as well as having improved performance on exome data compared to previous tools. We benchmarked the utility of SavvyCNV by comparing it to the current tools for calling CNVs in off-target regions in both targeted sequencing and whole exome sequencing (using a truth set derived from genome sequencing), and in on-target regions (using a truth set derived from MPLA). We then used SavvyCNV in a patient cohort tested with a small targeted gene panel (75 genes) to perform a genome-wide analysis to detect CNVs of clinical relevance.

## Results

### How much off-target read data is there?

For our small targeted panel [17] of 75 genes, 3.4 (SD 1.6) million reads are sequenced on average per sample. 55% (SD 10%) of these map to off-target regions of the genome. This gives a mean read depth in off-target regions of 0.065 (SD 0.044). In the exome samples that we used as a benchmarking data set there are an average of 76 (SD 20) million reads per sample with 20.3% (SD 6.6%) off-target, equating to mean read depth of 0.52 (SD 0.20) in off-target regions. This compares to a typical genome sequencing experiment where sufficient reads are sequenced to give >30X mean coverage across the genome.

### SavvyCNV can call off-target CNVs from targeted panels

To evaluate SavvyCNV's ability to call off-target CNVs accurately from targeted panel data we benchmarked its performance against a truth set of deletions and duplications generated using genome sequencing of the same samples (see Materials and Methods) and compared it to five other tools for calling CNVs: GATK gCNV [18], DeCON [19], EXCAVATOR2 [14], CNVkit [15], and CopywriteR [16]. To prevent bias due to software configuration tuning, we ran all six tools with multiple configurations, and plotted the best results for each tool on a precision-recall graph (Fig 1). The best recall (sensitivity) where precision is at least 50% is shown in Table A in S1 Text.

All tools called all of the CNVs larger than 5Mb (although not necessarily with precision of at least 50%), however only SavvyCNV did so without any false positive calls. All CNVs larger than 1Mb were called by SavvyCNV, GATK gCNV, and DeCON (all with precision less than 50%), although SavvyCNV called the most (97.6%) at a precision of at least 50% (as in Table A in S1 Text). For CNVs of any size, SavvyCNV had the highest recall (25.5%) with precision of at least 50%. For all three CNV size categories, SavvyCNV had the greatest detection power. SavvyCNV can call CNVs that are larger than 1Mb from off-target reads from a targeted panel with good recall (97.6%) and precision (78.8%).

Of the 68 CNVs detected by SavvyCNV (all sizes, precision at least 50%), 17 overlapped with targeted regions. This shows that while the increased read depth across a targeted region is used by SavvyCNV to help detect CNVs, most of the CNV detections are truly in off-target regions. Off-target calling is also able to map the boundaries of on-target CNVs where they extend outside of the targeted regions (see Fig A in S1 Text for an example of this).

We analysed the accuracy of the locations of the boundaries of the CNVs as detected by all six tools. SavvyCNV, GATK gCNV, and DeCON all had a similar accuracy, which scaled proportionately to the analysis bin size. With a bin size of 200kbp, all three tools found the CNV boundaries with a mean absolute error in location of 90kbp. Excavator2, CopywriteR, and CnvKit had higher location errors of 200kbp, 400kbp, and 600kbp respectively, although they detected very few CNVs and the results may not be reliable.

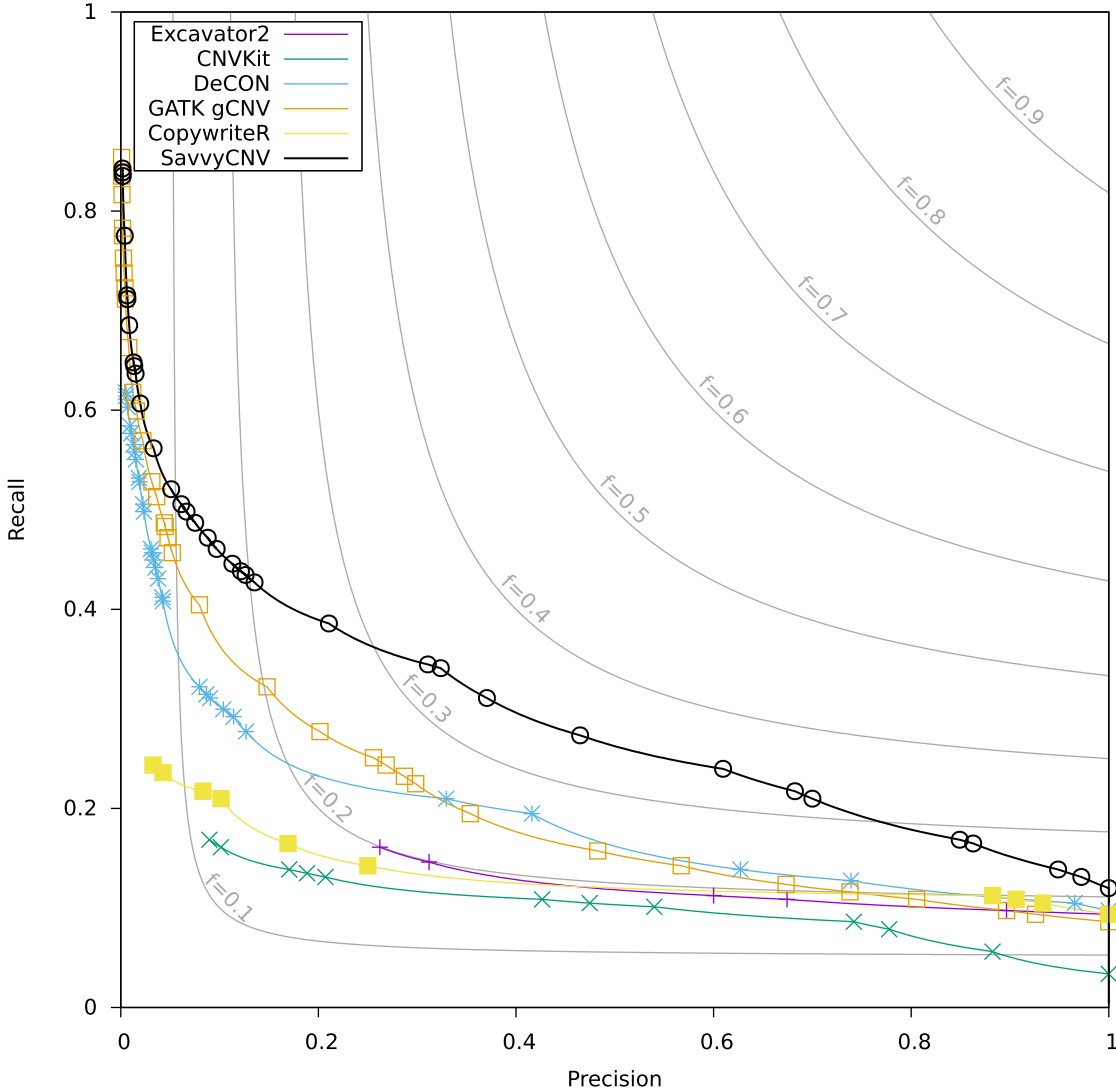

**Fig 1. Benchmarking off-target CNV calling from targeted panel data.** The data points on the plot are generated by a parameter sweep for each tool and show the precision and recall that can be achieved with each tool. The f statistic is the harmonic mean of precision and recall (see Materials and Methods for details).

We conducted a sensitivity analysis where an alternative CNV caller was used to call the truthset from the genome sequencing data and results were similar (see Fig B in S1 Text).

## SavvyCNV can call on-target CNVs from targeted panels

To evaluate the performance of SavvyCNV at calling CNVs from on-target data we used the ICR96 validation series [20] and compared its performance to GATK gCNV, DeCON, and CNVkit. ICR96 is a set of 96 samples sequenced using a small targeted sequencing panel (Tru-Sight Cancer Panel v2, 100 genes), with exon CNVs detected independently using MLPA (25 single-exon CNVs, 43 multi-exon CNVs, and 1752 normal copy number genes). SavvyCNV had the highest recall (for precision >= 50%) though GATK gCNV and DeCON also performed well—these 3 tools had a recall >95% (Table B in S1 Text). Precision can only be compared between tools if recall is identical. To give an example, GATK gCNV achieves its highest

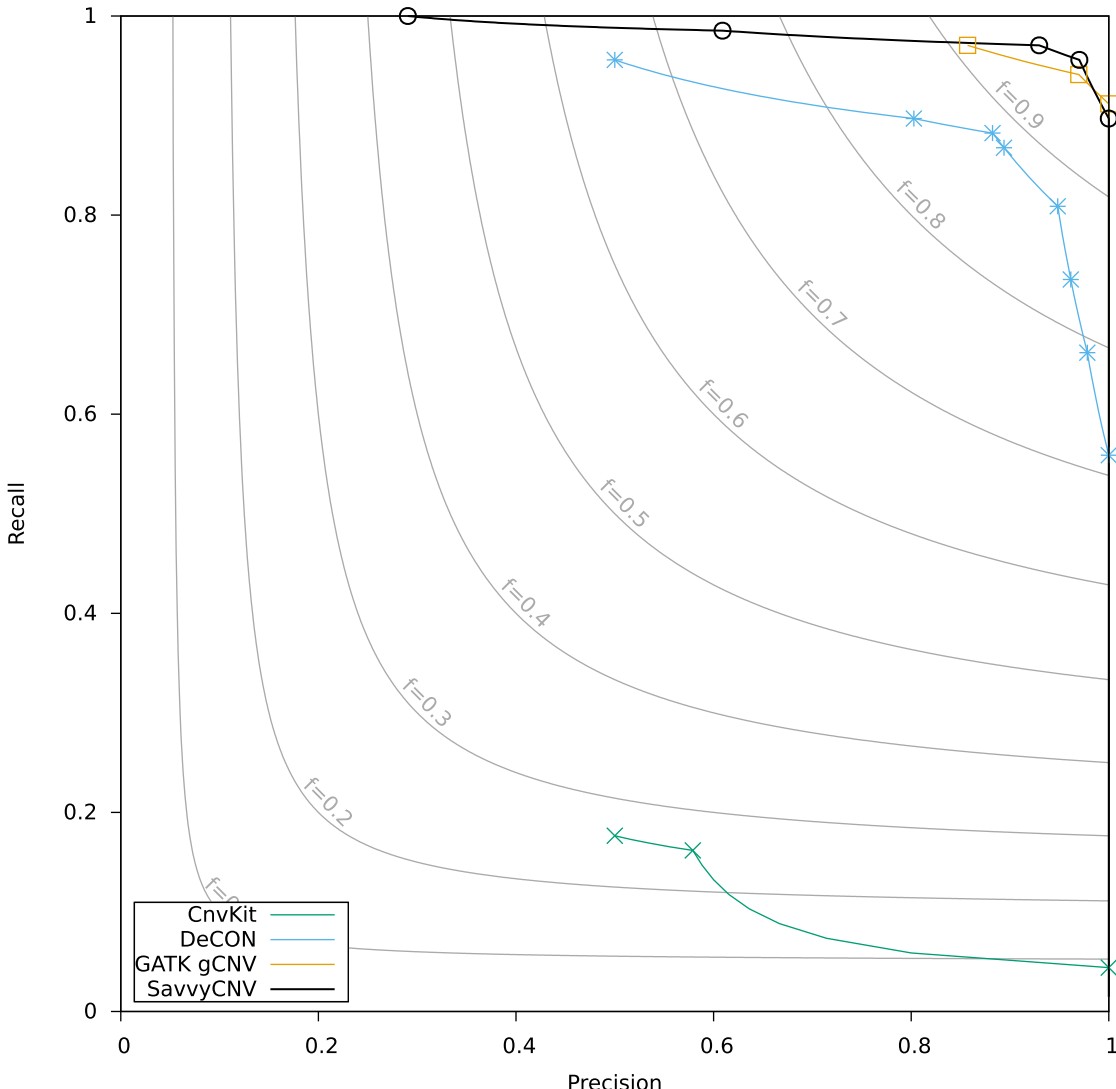

**Fig 2. Benchmarking on-target CNV calling from targeted panel data.** The data points on the plot are generated by a parameter sweep for each tool and show the precision and recall that can be achieved with each tool. The f statistic is the harmonic mean of precision and recall (see Materials and Methods for details).

recall of 97.1% with 85.7% precision, at the same level of recall SavvyCNV has a precision of 93.0% (this is shown in Fig 2). DeCON was the next-best performing tool after SavvyCNV and GATK gCNV while CnvKit did not call the majority of CNVs. Excavator2 did not run on this data set, and CopywriteR does not call on-target CNVs by design. Fig 2 shows the recall and precision of the four tools. SavvyCNV was the only tool capable of detecting all the CNVs although only with a precision of 29.1%.

Two of the CNVs within the ICR96 dataset cover less than a complete exon and have one breakpoint within the targeted region. These two CNVs are the hardest to detect by read-depth methods, as the read depth is only altered over a fraction of the exon area. Both CNVs are detected only by SavvyCNV, even when the highest sensitivity settings are used with the other CNV callers. If a structural variant caller is used to search for targeted breakpoints, then 7 of the 68 CNVs can be found in this data set. Adding a structural variant caller to the analysis

therefore increases the power to detect CNVs, even on targeted sequencing data. Of the remaining 61 CNVs, only SavvyCNV is capable of detecting them all (precision 59.2%) (Table B in S1 Text).

Multi-exon CNVs are easier to detect than single-exon CNVs. SavvyCNV, GATK gCNV, and DeCON can detect all 43 multi-exon CNVs, although only SavvyCNV and GATK gCNV did this with a precision of at least 50%.

## SavvyCNV can call off-target CNVs from exome data

To assess SavvyCNV's ability to call CNVs from off-target reads generated by exome sequencing we benchmarked it against a truth set (see Materials and Methods) and compared its performance to GATK gCNV, DeCON, EXCAVATOR2, CNVkit, and CopywriteR. The best recall where precision is at least 50% is shown in Table C in S1 Text for two different size categories, and recall/precision is shown in Fig 3 for all CNVs.

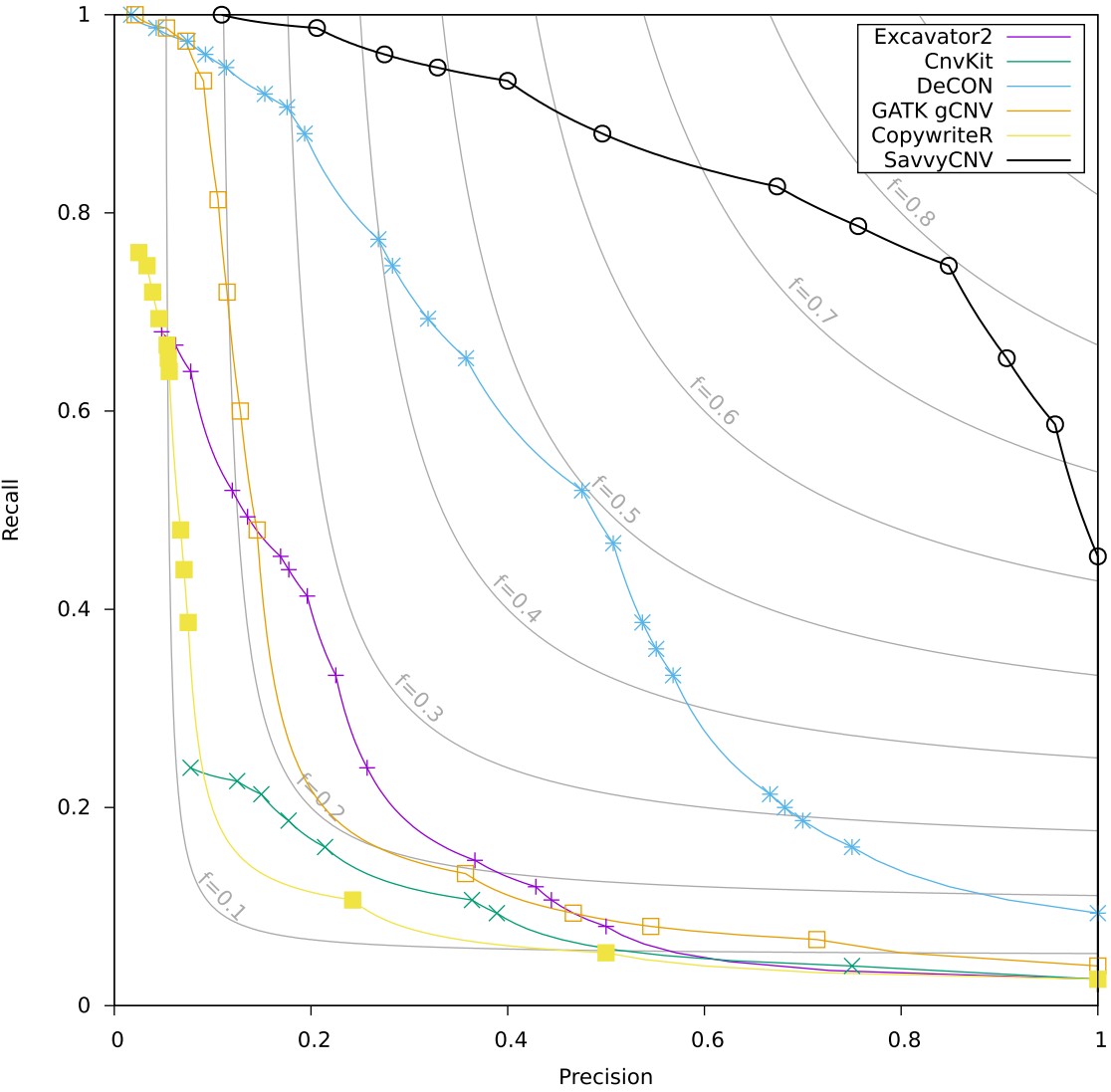

**Fig 3. Benchmarking off-target CNV calling from exome data.** The data points on the plot are generated by a parameter sweep for each tool and show the precision and recall that can be achieved with each tool. The f statistic is the harmonic mean of precision and recall (see Materials and Methods for details).

**Table 1. Clinically-relevant CNVs detected.**

| Row | CNV detected (GRCh37) | CNV size | Clinical confirmation | Reason for referral | Clinical implications |
|---|---|---|---|---|---|
| 1 | Chr8:6,800,000–11,800,000 deletion | 5Mb | Deletion includes *GATA4*, causative of the patient's neonatal diabetes and additional features. | Diabetes | Genetic diagnosis of monogenic diabetes [21]. |
| 2 | Chr8:8,000,000–10,400,000 duplication 8:10,600,000–12,000,000 deletion | 2.4Mb and 1.4Mb | Deletion includes *GATA4*, causative of the patient's neonatal diabetes and additional features. | Diabetes | Genetic diagnosis of monogenic diabetes [21]. |
| 3 | Chr18:19,400,000–21,800,000 deletion | 2.4Mb | Deletion includes *GATA6*, causative of the patient's neonatal diabetes and additional features. | Diabetes | Genetic diagnosis of monogenic diabetes [22]. |
| 4 | Chr21:14,400,000–48,200,000 duplication | Chromosome | Patient known to have Down Syndrome at referral. | Hyperinsulinism | Confirms the diagnosis of Down syndrome. |
| 5 | Chr22:18,800,000–21,600,000 deletion | 1.8Mb | Confirmed by array CGH. | Diabetes | Provided the diagnosis of DiGeorge syndrome. |
| 6 | ChrX:0–155,400,000 duplication | Chromosome | Patient known to have XXX syndrome at referral. | Diabetes | Confirms the diagnosis of XXX syndrome. |

SavvyCNV was the best performing tool on this data set, able to call 86.7% of the CNVs with at least 50% precision, while the next best tool (DeCON) called 46.7% of CNVs with at least 50% precision. The chief difference between the performances of the tools is SavvyCNV's ability to call CNVs smaller than 200kpb. SavvyCNV is able to call an additional 30 CNVs that are smaller than 200kpb at $\geq$ 50% precision while GATK gCNV, EXCAVATOR2, CNVkit, and CopywriteR call no true CNVs smaller than 200kB, and DeCON calls 10.

We conducted a sensitivity analysis where an alternative CNV caller was used to call the truthset from the genome sequencing data and results were similar (see Fig C in S1 Text).

## SavvyCNV can detect clinically relevant CNVs

Having validated the ability of SavvyCNV to call CNVs from off-target reads we proceeded to screen for CNVs in our cohort of targeted panel samples from patients referred for genetic testing to identify the cause of their diabetes or hyperinsulinism [17]. We were able to detect 6 clinically relevant CNVs both within and outside of the targeted regions (Table 1). Of these, 3 provided a new genetic diagnosis for diabetes/hyperinsulinism (rows 1–3 in Table 1), providing information which will guide clinical management and allow accurate counselling on recurrence risk in family members and future offspring. The remaining 3 CNVs (rows 4–6 in Table 1) confirmed clinically-reported diagnoses unrelated to the diabetes/hyperinsulinism. These findings demonstrate the ability of SavvyCNV to detect clinically relevant CNVs and aneuploidies from off-target data from a small targeted panel.

## Discussion

### SavvyCNV can detect CNVs genome-wide from off-target reads

We benchmarked SavvyCNV on its ability to call germline off-target and on-target CNVs from targeted panel and exome sequencing data. This new tool outperformed five existing tools, three of which (CNVkit [15], Excavator2 [14], and CopywriteR [16]) were specifically designed to call off-target CNVs. GATK gCNV performed similarly to SavvyCNV in the on-target (ICR96) analysis. However, SavvyCNV considerably outperforms all other tools in the off-target analyses.

SavvyCNV finds the greatest number of true positive CNVs in all data sets while other tools did not call certain CNVs. For example, the two partial exon CNVs in the on-target (ICR96)

data set are detected only by SavvyCNV. This is likely because of the improved error correction and error modelling that is incorporated into SavvyCNV over existing tools. SavvyCNV uses singular vector decomposition to reduce noise. CNVkit, EXCAVATOR2 and CopywriteR only correct for GC content, while GATK gCNV uses Bayesian principle component analysis (https://www.broadinstitute.org/videos/scalable-bayesian-model-copy-number-variation-bayesian-pca), and DeCON uses sample matching (it searches for samples in the control set that have a similar noise profile). Unlike the other tools tested, CopywriteR does not normalise against other samples but excludes on-target reads to make read counts representative of the true copy number. CNVkit was primarily designed for somatic CNV calling and thus it is perhaps not surprising that when used for germline calling it demonstrates limited performance.

SavvyCNV had a higher precision than other tools when calling off-target CNVs, an important consideration in diagnostic and research laboratories as if false positives are reduced, fewer CNVs will require orthogonal testing to identify the true positive results. Many of the false positives produced by DeCON, CNVKit, and EXCAVATOR2 have a read depth ratio close to 1, where a true full deletion should be close to 0.5 and a true full duplication should be close to 1.5. This indicates that these tools are picking up either mosaic CNVs or noise. The prior probability is overwhelmingly that these are noise. This is why the default for SavvyCNV and GATK gCNV is to call only non-mosaic CNVs as this hugely reduces the number of false positives called. Mosaic CNV calling can be enabled in SavvyCNV for projects where it is applicable.

SavvyCNV can also be used to call CNVs from on-target sequencing and, while it is not demonstrated in this study, it can be used to call CNVs from genome sequencing data and has already been used to identify a novel disease-causing deletion [23].

## Estimates of precision and recall rely on the quality of the truth set

On-target CNV calling from a targeted panel was tested on the ICR96 data set in which the truth set was verified by MLPA. The truth sets for the off-target CNV calling from targeted panel and exome sequence data were generated from CNV calls from genome sequencing data. Genome sequencing has a much higher coverage than that generated from only off-target reads which allows CNVs to be called more accurately enabling them to be used as a truth set. GenomeStrip [24] was used to call the truth set as it was designed to call CNVs from genome data and was not one of the tools under examination in this study. We also conducted a sensitivity analysis where Canvas was used to call the truthset and results were similar. However, it is possible that there could be some false positive and negative calls in the truth set. This would lower the precision and recall of the tools under examination but should not bias the results in favour of a particular tool. Furthermore, as the majority of the truthset was called using a read depth method, and SavvyCNV and the tools it was compared to also use read depth methods, it is possible that they would share technical artifacts which could artificially inflate estimates of performance.

## Sensitivity depends on the size of the CNV

Smaller CNVs are harder for all software to detect. For all tools tested the larger the CNV the better the precision and recall, however SavvyCNV performs better than the other tools tested. SavvyCNV detects CNVs larger than 1Mb with 100% recall in off-target data from both targeted panel and exome data.

Another important factor determining performance is the number of off-target reads. The number of off-target reads will vary depending on the depth of sequencing, size of panel and the particular capture method used. We have created an additional tool, CoverageOffTarget

(available at https://github.com/rdemolgen/SavvySuite), which calculates the number of off-target reads in a set of samples.

## CNV calling can be optimised for precision or recall by adjusting configuration

When calling CNVs, precision and recall are a trade-off; high recall will maximise the number of true CNVs that are called, with the consequence that it also reduces precision resulting in a large number of false positive CNV calls. Precision and recall can be adjusted using the bin size and transition probability parameters—for full documentation on how to use the software see the github page: https://github.com/rdemolgen/SavvySuite. The CoverageOffTarget tool can be run on the set of samples to be analysed to provide a recommendation for an appropriate bin size.

Different precision levels are appropriate in different situations, influenced both by the experimental methodology and the aims of the project. When calling CNVs on-target on a small gene panel there will be fewer false positive calls generated due to the smaller target area thus it may be preferable to adjust settings to enable a higher recall at the cost of a lower precision. This could also be true in a clinical context where the most important aim is to not miss a true causative variant. In contrast, when calling CNVs genome-wide in a gene-agnostic approach such as exome or genome sequencing, a higher precision is likely to be desirable to avoid generating an unmanageably long list of CNVs. A large bin size may also be preferable when the aim is to detect large CNVs as this will improve the precision for detecting these with the trade-off of potentially missing smaller CNVs. The user can choose their preferred settings for SavvyCNV for different project requirements.

## Off-target CNV calling is 'free' data and increases diagnostic yield

SavvyCNV utilises data already generated by targeted panel and exome tests. These tests are carried out in order to detect single nucleotide variants and small insertions or deletions (<50 base pairs). In some laboratories CNVs are also detected within the targeted regions using CNV calling software while other laboratories use array-CGH or MLPA to detect CNVs. Using SavvyCNV allows CNVs to be detected not just within the targeted regions but allows genome-wide CNV calling. This will provide a genetic diagnosis for more patients, increasing the diagnostic yield of these tests. We have demonstrated the ability to find relevant genetic diagnoses using off-target CNV calling from our small targeted panel. Existing data can be reanalysed with our method to reveal additional CNVs. As an illustration of this, two of the CNVs in the ICR96 data set were found to actually be large CNVs (15Mb and 56Mb), which may have clinical implications beyond the targeted gene.

SavvyCNV calls germline CNVs from off-target reads from exomes and small targeted gene panels with high precision and recall, and performs better than existing tools including those designed for off-target CNV calling. Calling CNVs from off-target reads is exploiting 'free' data to increase the diagnostic yield of targeted panel and exome sequencing tests and reveal important biological findings.

## Materials and methods

### Ethics statement

For the patients referred to the molecular genetics department at the Royal Devon and Exeter Hospital for genetic testing, informed written consent was obtained from the probands or their parents/guardians, and the study was approved by the North Wales ethics committee. The study was conducted in accordance with the Declaration of Helsinki.

## Overview of how SavvyCNV calls CNVs

SavvyCNV and the commands to run it are freely available from https://github.com/rdemolgen/SavvySuite.

SavvyCNV calls CNVs by looking at read depth over the genome. The genome is split into bins and each bin is assessed for statistical divergence from normal copy number. The bin size is user specified. If there is targeted sequencing with three million reads and approximately 50% off-target reads, then a bin size of 200kpb is appropriate.

SavvyCNV normalises the read depths by first dividing the read count by the mean relative read depth of the sample across all genomic locations, and then subsequently dividing the read count by the mean read depth of the genomic location across all samples. SavvyCNV then uses singular vector decomposition (SVD) to reduce noise. This identifies biases common to multiple samples, which can be caused by differences in sample handling or chemistry. The SVD is performed on the logarithm of the normalised read depth, and by default the first five singular vectors are discarded. Fig D in S1 Text shows the effect of using SVD on precision and recall. SavvyCNV estimates the error in each genomic location in each sample by modelling the error using the average error in each sample across all genomic locations and the average error in each genomic location across all samples. The total number of reads available is also analysed, and the error estimation is increased if it would be lower than the error calculated using the Poisson distribution. The effect on CNV calling accuracy is shown in Fig E in S1 Text while the design of the error modelling is described in Fig F in S1 Text. The error estimate is used to determine whether the read depth in a genomic location is significantly outside the range expected for normal copy number. The significance probability of the read depth representing a deletion, duplication, or normal dosage is calculated using an approximation of the cumulative normal distribution ($P(d < = 0.5)$ for deletions, $P(d > = 1.5)$ for duplications, and $P(d < = 1.0)$ or $P(d > = 1.0)$ for normal dosage). These probabilities are used in a hidden Markov model (HMM) with three states (deletion, duplication, and normal) to identify CNVs.

SavvyCNV assumes CNVs are non-mosaic by default. The HMM state probability calculations assume that the relative dosage is either $< = 0.5$, 1, or $> = 1.5$. Dosage levels must cross the mid-point between 1 and 0.5 or 1.5 (so the probability of a deletion or duplication is calculated as greater than the probability of normal dosage) before they become evidence of a CNV. This increases specificity at the cost of being able to detect mosaic CNVs (see Fig G in S1 Text). Mosaic CNV calling can be enabled in SavvyCNV for projects where it is applicable, which changes the probability calculations to use just the probability of the read depth being not 1.

## Computational requirements

Run time is proportional to the number of active bins in the analysis, and proportional to the number of samples to the power of 2.3. Calling on-target CNVs on 150 exomes takes around half an hour, but 300 exomes would take five times as long. Calling off-target CNVs on 1700 targeted samples with a bin size of 200kpb takes around 4 hours.

If only a small number of samples are available, then CNV detection performance is lower. Fig H in S1 Text shows how the CNV detection ability of SavvyCNV depends on the number of samples that are analysed in a single batch.

If a large number of samples must be processed, then they can be split into groups and processed separately. Additional software is provided alongside SavvyCNV that allows a selection of similar samples to be selected from a large pool to use as a control for a set of samples, to reduce run time. RAM usage depends on the bin size and the number of samples. Calling on-target CNVs on 1700 samples with a bin size of 400 base-pairs requires 170GB of RAM, and calling off-target CNVs on 1700 samples with a bin size of 200kpb requires around 1GB of RAM.

### Targeted panel data

2591 patients were referred to the molecular genetics department at the Royal Devon and Exeter Hospital for genetic testing for maturity onset diabetes of the young (MODY), neonatal diabetes (NDM) or hyperinsulinemic hypoglycemia (HH). Samples were sequenced on a targeted gene panel test for monogenic diabetes and HH using a custom Agilent SureSelect panel of 75 genes, targeting 200kpb and obtaining 3.4 million reads on average per sample (standard deviation 1.6 million)[17], using an Illumina HiSeq 2500 or an Illumina NextSeq 500. Based on the GATK best practice guidelines [18] reads were aligned to the hg19/GRCh37 human reference genome with BWA mem [25], duplicates were removed using Picard (https://broadinstitute.github.io/picard/) and GATK IndelRealigner was used for local re-alignment. CNV analysis was carried out on these BAM files.

### Exome sequencing data

Following testing using the targeted panel, samples from 86 patients underwent exome sequencing with Agilent SureSelect Whole Exome versions 1, 3, 4, and 5, obtaining 76 million reads on average (standard deviation 20 million). Sequencing, alignment and variant calling was as above for the targeted panel.

### Truth set for targeted and exome data

The truth set was based on genome sequencing of the same samples. 170 of the targeted panel samples and 42 of the exome samples were subsequently genome sequenced on an Illumina HiSeq 2500 or an Illumina HiSeq X10. These were used to create a truth set of CNVs for testing the off-target CNV calling from targeted panel or exome data. The CNVs in the truth set were called by GenomeStrip [24] from the genome sequencing data. In order to remove false positive calls CNVs were filtered based on their allele balance ratios–whether the allele balance of the variants within the called CNV was consistent with it being a true call. We used the X chromosome in males to calibrate the expected allele ratio for a deletion and used the allele ratio of normal, two copy regions to evaluate if the allele ratio for duplications fell above that. In addition 37 CNVs were added to the targeted panel truth set as they were validated by other methods such as MLPA or were aneuploidies reported by the clinician at time of referral for genetic testing. There were 45 duplications and 30 deletions tested in the exomes, and 171 duplications and 96 deletions tested in the targeted samples.

We conducted a sensitivity analysis where Canvas [26] was used to call the truthsets from the genome sequencing data. The ICR96 data set [20] was used to benchmark on-target CNV calling. This data set consists of 96 samples sequenced on a targeted panel where the truth set of CNVs is based on 68 positive and 1752 negative MLPA tests.

### Calling clinically-relevant CNVs

The remaining 2479 targeted panel samples from unsolved patients with MODY, NDM and HH were analysed with SavvyCNV to look for off-target CNVs which might explain their phenotype. For clinical evidence of the CNVs, see Table 1.

### CNV tool comparisons

A CNV was deemed to have been detected if there was any overlap between the call made by the CNV caller and the truthset. To ensure a fair comparison between the different tools, for each data set all tools were run with a variety of configurations. The size of genomic regions that were analysed was varied for all six tools (targeted panel: 150kbp to 300kbp or 50kbp to

2Mbp for CopywriteR; exomes: 6kbp to 50kbp or 20kbp to 2Mbp for CopywriteR; ICR96: 200bp to 600bp). The hidden Markov model transition probability was varied for DeCON and SavvyCNV ($10^{-10}$ to 0.1). All six tools provide quality metrics for the CNV calls. These metrics were used to filter the CNV calls to reject false positive calls and retain true positive calls. All possible quality cut-off values were tried. The best precision achieved for each possible recall was then selected for each tool from all the generated results, and plotted in precision-recall graphs. GATK gCNV produces a probability of CNV in each individual genomic location, but does not estimate the boundaries of detected CNVs. The output was processed to group together locations with probable CNVs to determine the size and boundaries of the CNVs, in order to produce results that were comparable to the other CNV callers. EXCAVATOR2 [14] did not run on the ICR96 data set—we contacted the authors of the tool but did not receive a response. CopywriteR was not run on the ICR96 data set, as it is not designed to use on-target data. The code used to run the benchmarking comparisons is available at https://github.com/exeter-matthew-wakeling/SavvyCNV_benchmarking.

## Statistics

We defined recall as the percentage of true positive CNVs that were found by the tool. We defined precision as the percentage of the total CNVs called by the tool that were true. Several figures use the f statistic to compare tools; this is the harmonic mean of precision and recall.

## Supporting information

**S1 Text. Supplementary information. Table A**. Benchmarking off-target CNV calling from targeted panel data. The table shows the performance of the different CNV calling software based on the size of the CNV. The tools were run with multiple different parameters. For this comparison, we have selected the configuration for each tool that provides the highest recall with a precision of at least 50%. More variants may be detected by each tool with different configuration, but with precision less than 50%. **Table B**. Benchmarking on-target CNV calling from the ICR96 targeted panel data. The table shows the performance of the different CNV calling software based on the size of the CNV. The tools were run with multiple different parameters. For this comparison, we have selected the configuration for each tool that provides the highest recall with a precision of at least 50%. **Table C**. Benchmarking on-target CNV calling from the exome data. The table shows the performance of the different CNV calling software based on the size of the CNV. The tools were run with multiple different parameters. For this comparison, we have selected the configuration for each tool that provides the highest recall with a precision of at least 50%. **Fig A**. An example of a single large CNV called by SavvyCNV in chromosome 8. The CNV is a heterozygous deletion of 8:6,800,000–12,400,000, which is a 5.6Mbp deletion. The CNV analysis uses a bin size of 200kbp, so the CNV covers 28 bins. The normalised read depth of each bin is shown as an error bar, where the error is the estimated error calculated by SavvyCNV. Bins either side of the deletion have a normal normalised read depth near 1, whereas bins inside the deletion have a normalised read depth near 0.5. This CNV overlaps a targeted gene in the sequencing capture, and this targeted data covers two separate bins, which are marked. Note that the error estimate for these two bins is slightly smaller than the other bins, partly because of the increased read count contributed by the targeted region of the capture, however the difference is small–the majority of evidence for this CNV comes from off-target data. Some of the bins have a large estimated error–this is caused by large tandem repeat regions which mean the read depth is highly variable between samples. **Fig B**. Benchmarking off-target CNV calling from targeted panel data. This figure shows the equivalent of Fig 1, but using a truth set called using Canvas on whole genome sequenced

samples. The truth set used in Fig 1 was produced by analysing whole genome samples with GenomeStrip2 and filtered using variant allele fraction data. The data points on the plot are generated by a parameter sweep for each tool and show the precision and recall that can be achieved with each tool. The f statistic is the harmonic mean of precision and recall (see Materials and Methods for details). **Fig C**. Benchmarking off-target CNV calling from exome data. This figure shows the equivalent of Fig 3, but using a truth set called using Canvas on whole genome sequenced samples. The truth set used in Fig 3 was produced by analysing whole genome samples with GenomeStrip2 and filtered using variant allele fraction data. The data points on the plot are generated by a parameter sweep for each tool and show the precision and recall that can be achieved with each tool. The f statistic is the harmonic mean of precision and recall (see Materials and Methods for details). **Fig D**. Shows the improvement in precision/recall due to the error correction strategy used by SavvyCNV, for all three data sets. By default, SavvyCNV uses singular vector decomposition (SVD), which identifies biases common to multiple samples, which can be caused by differences in sample handling or chemistry. The "normal" line shows the default configuration, while the "No SVD" line is with SVD switched off. **Fig E**. Shows the improvement in precision/recall due to the error modelling strategy used by SavvyCNV. By default, SavvyCNV estimates the error in normalised read depth in each genomic location for a sample by calculating the standard deviation for that location across all samples, then scaling by the standard deviation for that sample across all locations. An alternative is to add the two error values. Other software assumes that the error is Poisson in nature, and can therefore be calculated from the read depth. The error estimate is used to determine whether the read depth in a genomic location is significantly outside the range expected for normal copy number. In reality, the actual error is larger than the Poisson method in some genomic locations. Modelling the error allows SavvyCNV to avoid making false CNV calls in these highly variable regions. **Fig F**. Shows the error estimation accuracy of three strategies investigated during the development of SavvyCNV. The CNV detection ability of these three strategies is shown in Fig E in S1 Text. SavvyCNV was used to analyse the read depths of around a thousand targeted samples with a bin size of 200kbp. The estimated error was calculated using the three strategies and listed alongside the actual normalised read depth value for each bin in each sample, giving a total of 32 million data points. These data points were then grouped into 100 sets with similar estimated error, and the standard deviation of the normalised read depth was calculated—this is the actual error. The Poisson error model calculates the estimated error by using the number of reads in the analysis bin, and it usually underestimates the actual error, however it does represent a theoretical minimum random error that the normalised read depth could have. The additive error is formed by adding the standard deviation of the genomic location across all samples with the standard deviation of the sample across all genomic locations. This usually overestimates the error, but this cannot be corrected by scaling. The normal error model multiplies the two error estimates together and divides by the mean error of all samples, and is highly effective at estimating the error in the normalised read depth. As a check, if this calculation yields an error lower than the Poisson error, then the Poisson error is used instead. **Fig G**. Shows the improvement in precision/recall due to the non-mosaic assumption used by SavvyCNV. By default, SavvyCNV assumes that CNVs are not mosaic, although this is a configurable setting. This allows it to reduce the number of false positive CNV calls. In mosaic mode (and DeCON, Excavator2, and CNVKit), a CNV call is produced when the normalised read depth is significantly away from 1.0. In SavvyCNV's default mode (and GATK gCNV), the read depth must also be closer to 0.5 or 1.5 (representing a whole heterozygous deletion or duplication) than 1.0 in order for a CNV to be called. **Fig H**. Shows how the CNV detection ability of SavvyCNV depends on the number of samples that are analysed in a single batch. SavvyCNV detects regions where the read depth of a sample is

higher or lower than expected, and so it must have other samples to compare the test sample to. A larger number of available samples improves the ability of SavvyCNV to correct for noise and determine the expected read depth for each sample, to better detect deviations from that expected read depth. We divided our set of targeted sequencing samples into randomly assigned batches sized between 7 and 400 samples, and ran SavvyCNV with multiple configuration settings as described for the main experiment that produced Fig 1 for each group size. The maximum f statistic was calculated for each group size. This process was repeated 25 times with randomly-assigned groups. This shows that the detection power of SavvyCNV for this set of samples is lower when fewer samples are analysed. The power to detect all CNVs is fully reached when there are at least 50 samples analysed. The power to detect larger CNVs continues to increase with larger sample batches up to a maximum with 200 in each batch. (DOCX)

## Author Contributions

**Conceptualization:** Matthew N. Wakeling.

**Data curation:** Elisa De Franco, Matthew B. Johnson, Kashyap A. Patel, Sarah E. Flanagan.

**Formal analysis:** Matthew N. Wakeling.

**Funding acquisition:** Elisa De Franco, Matthew B. Johnson, Kashyap A. Patel, Sian Ellard, Michael N. Weedon, Sarah E. Flanagan.

**Methodology:** Thomas W. Laver, Michael N. Weedon, Matthew N. Wakeling.

**Project administration:** Thomas W. Laver.

**Software:** Matthew N. Wakeling.

**Supervision:** Sian Ellard, Sarah E. Flanagan.

**Validation:** Thomas W. Laver, Michael N. Weedon, Matthew N. Wakeling.

**Visualization:** Matthew N. Wakeling.

**Writing – original draft:** Thomas W. Laver, Matthew N. Wakeling.

**Writing – review & editing:** Elisa De Franco, Matthew B. Johnson, Kashyap A. Patel, Sian Ellard, Michael N. Weedon, Sarah E. Flanagan.

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
