## [Decision Letter · Decision Letter 0]

14 Oct 2021

Dear Dr Laver,

Thank you very much for submitting your manuscript "SavvyCNV: genome-wide CNV calling from off-target reads" for consideration at PLOS Computational Biology.

As with all papers reviewed by the journal, your manuscript was reviewed by members of the editorial board and by several independent reviewers. In light of the reviews (below this email), we would like to invite the resubmission of a significantly-revised version that takes into account the reviewers' comments.

We cannot make any decision about publication until we have seen the revised manuscript and your response to the reviewers' comments. Your revised manuscript is also likely to be sent to reviewers for further evaluation.

Sincerely,

Mihaela Pertea

Software Editor

PLOS Computational Biology

Jason A. Papin

Editor-in-Chief

PLOS Computational Biology

Reviewer's Responses to Questions

**Comments to the Authors:**

Reviewer #1: The paper by Laver et al presents SavvyCNV, a tool for CNV calling from hybrid capture sequencing data which can make use of off-targets reads. While plenty of tools have been developed for CNV calling from exome and panel sequencing data, CNV calling cannot be considered as a solved problem: concordance between calls from different tools is low, and false negatives and false positives are common. Hence there is still a high (and with the increasing use of panel and exome sequencing for diagnostics growing) need for improved CNV calling methods.

The authors benchmark SavvyCNV against five other tools in three settings: off-target calling from exome data, off-target calling from panel data, and on-target calling from panel data. For benchmarking calling using off-target reads, an in-house data set with panel sequencing data from 2591 patients and exome sequencing data from 86 patients. The truth set for these benchmarks was obtained from deep whole genome sequencing of 170 panel-sequenced and 42 exome-sequenced samples. As truth, the authors considered CNV calls obtained by GenomeStrip 2.0 from this genome sequencing data. On-target calling was benchmarked using the previously published IRC96 dataset. For all tools, parameter sweep was performed and performance was evaluated in precision recall curves. SavvyCNV had the best overall performance in all shown comparisons. In the supplement, the authors add results which demonstrate the impact of three approaches used in the CNV calling process of SavvyCNV.

Overall, the authors have performed a comprehensive comparison to demonstrate the superiority of their tool in the tested scenarios. Based on these results, SavvyCNV appears to be a considerable improvement compared to the other callers included in the benchmark and thus might be of high value for a broader community.

However, there are still some issues which should be addressed:

Major issues:

1. The truth set for off-target calling can be expected to contain false negatives and false positives. The statement of the authors, that these will affect all tools to the same amount is not necessarily true: tools with a similar error profile as GenomeStrip 2.0 will have an advantage in this benchmark and might even appear to have a better performance than a perfect caller. Although one could hypothesize that calling from whole genome data is sufficiently more powerful to make the errors of GenomeStrip negligible, an additional comparison using an independent truth set would make the manuscript stronger. The authors could for example use the recently compiled validation set on NA12878 published by Gordeeva et al. (Scientific Reports 2021).

2. The methods descriptions in the manuscript are incomplete. This affects both the savvyCNV algorithm and the comparison. For example, it is not described at all how savvyCNV performs calling. There are only rather general descriptions of noise handling and the error model, but no in-depth methods; obviously, SavvyCNV employs a hidden Markov Model (mentioned in the CNV tools comparison section), but there is no description how it is implemented. Similarly, more details should be given about the CNV calling benchmark. How exactly have CNVs been compared? Was a CNV call considered as correct if it was partially overlapping with a reference call but had shifted start and/or end coordinates? Was the classification just gain or loss, or was there a discrimination of hemi- and homozygous deletions and different levels of gains.

3. More information should be included to help potential users to decide if savvyCNV could be the right tool for their application. In particular, it would be important to know how many samples are needed for an analysis and how similar these samples have to be. Many users won’t have access to such a big collection of samples as used here for exome and in particular panel off-target calling.

4. The code and in particular the exact parameters used for each tool in the benchmark should be provided.

Minor issues:

1. Page 11, discussing mosaic or random noise. I agree that it is very likely noise, but most likely not random.

2. CNVkit was primarily designed for somatic CNV calling, and the documentation mentions that when used for germline calling it can be expected to show limited performance. This should be briefly added to the discussion.

3. It seems that SavvyCNV has been developed and evaluated for germline CNV calling and not for somatic CNV calling. This should be mentioned in the manuscript; it could even be added to the title.

Reviewer #2: # Overview

Detection of CNVs from targeted sequencing and exome data is an important problem in clinical

genetics and there have been a wide range of tools published over the last decade. As noted

by the authors however, most of these tools achieve mediocre precision when applied at acceptable

sensitivity for clinical use, and further they typically can only operate well over the specific

genes targeted by the specific assay, making it impossible to characterise the real span of

larger genomic events even when they are detected.

SavvyCNV offers an exciting addition to this ecosystem of tools because it enables detection of

larger events completely outside targeted regions and characterisation of the size and breakpoints

of events that would otherwise be detected only within targeted regions. This is a valuable

contribution and even while other tools do exist, SavvyCNV bring several novel additions to the

process along with a high quality, well performing implementation.

In addition to reviewing the manuscript, I evaluated SavvyCNV on my own data. Although it required

several iterations of tuning the parameters of the software, after doing that I

was impressed with the speed and accuracy. It detected all known multi-exon CNVs and

nearly all or all single-exon CNVs, which depended on the configuration

parameters I chose. The software was easy to download, compile and run : it was

refreshing to see a pure Java implementation with minimal dependencies,

making it highly portable and performant.

Despite my good impressions of the software, I think the manuscript itself

needs some additions before it is fit for scientific publication. The problems

include insufficient description of methods, lack of clarity around terms used,

missing essential details for reproducibility and failure to acknowledge

significant weaknesses and limitations.

I have added specific notes below about areas that I think require addressing.

# General Comments

- Overall it is a bit confusing that evaluation of off-target calling is

combined with with on-target calling in various respects.

- When referring to "off target" CNVs, does this imply that there are no target regions

overlapping the true CNV? Or only that off-target evidence is the primary signal for detection?

- If on-target regions are included, are the read counts included in SavvyCNV's

algorithm? How does this affect the statistics when some bins contain spikes of very

high coverage while others only contain low level coverage? It seems to me this would

heavily distort, eg: the noise estimation which utilises standard deviation averaged

across all the bins.

I note that the software does not appear to take the target regions of the genome as

a parameter; therefore it seems that in regular use both targeted regions and non-targeted

regions would be considered by the tool together.

This leads to a contradiction: the primary claim of the manuscript is that SavvyCNV utilises

off-target reads, but in reality it appears that most of its detections would be aided by

at least *some* on target reads. To substantiate that off-target reads are really being utilised

I think there needs to be more detail given about (a) whether the results shown actually

overlap on-target reads or not and (b) whether the off-target reads aided in characterisation

of the true size / location of the CNV.

Further explanation around all of these issues is required.

# Specific Comments

> CNV calling can be optimised for precision or recall by adjusting configuration

The section describes that SavvyCNV can be optimised for different requirements, however

it does not present any details on how this is achieved. A summary of which parameters

can be tuned and some idea of methodology of approach for doing so should be included.

Is it possible without a dedicated truth set for the data under

consideration? since each different type of data needs potentially different parameter values,

it is not clear how a user can go about tuning these parameters in practice.

## Results

> How much off-target read data is there?

How well SavvyCNV performs is critically dependent on how many off target reads

there are. However this is a highly variable aspect of sequencing and capture

methods. The manuscript should ideally give a bit more background explanation

that off-target reads are influenced by the chemistry and laboratory process

and may vary significantly between different contexts - this to some extent is an important

limitation of the method and that should be acknowledged. It would be ideal to

give a table showing a variety of exome captures and targeted panels with

different chemistries and number of genes. Since this parameter is critical to

operation of SavvyCNV, it would seem useful if there is some kind of QC

statistic that is produced as part of an analysis that can inform the user of

the abundance of off target reads that were observed.

> SavvyCNV can detect clinically relevant CNVs

In Table 1, presumably in some of these cases the span of the CNVs detected include

some of the target regions of the panel (for example, the whole chrX aneuploidy). In these

cases it should be clarified to what extent the off-target detection was aided by on-target

reads. Presumably, SavvyCNV has enabled characterisation of the size of these events well

beyond what would possible from the target regions alone. However, to substantiate this,

it would be good for the results to include details showing how accurately SavvyCNV

characterised the size / location of the CNV compared to orthogonal validation data for

the events.

# Assessment of Off-Target CNV Calling

The truth set was constructed using GenomeSTRiP. This seems a surprising choice given that there

are many more well established whole genome sequencing CNV and SV callers now and GenomeSTRIP is

designed for population scale data. A key issue with this approach is that GenomeSTRiP is also

heavily read depth based, so it is susceptible to many of the same artefactual biases and could

mean that many calls in the "truth" set are shared false positives between GenomeSTRiP and

SavvyCNV. Similarly, many regions where GenomeSTRiP is unable to call CNVs will also apply

to SavvyCNV, so use of GenomeSTRiP is questionable for characterising the false negative rate.

While text acknowledges these issues in a minor way, this is somewhat buried and insufficiently

highlighted (eg: in the conclusions).

# Assessment of On-Target CNV Calling

- The manuscript doesn't state how much overlap was required for a CNV to be considered as successfully

detected. I think this is quite important in the case of SavvyCNV, as it primarily detects larger

events from off-target reads, but if these are only partially identified or identified in a

fragmented way then it would affect the usability in many settings.

# Materials and Methods

> Overview of how SavvyCNV calls CNVs

The overview is extremely brief and leaves many critical details out. For example, it is clear from the

source code and alluded to in other parts of the manuscript that SavvyCNV applies a hidden markov model

style segmentation algorithm using viterbi optimisation. However the overview does not include any mention

of this part of the algorithm. Further, within the HMM the likelihood is evaluated by use of a

normal distribution to model state probabilities, ie: reflecting a normal assumption for the distribution

of read counts. These are commonly examined properties of CNV detection algorithms, and certain limitations

arise from their characteristics (for example, utilising a normal model usually performs suboptimally for

low read counts, and HMM based methods often have issues at the start / end points of the model because a

neutral state is assumed as a starting point).

These details should be explained as important elements of the algorithm.

> Dosage levels must cross the mid-point between 1 and 0.5 or 1.5 before they become evidence of a CNV.

This increases sensitivity

It doesn't seem possible that this could increase sensitivity because it acts purely to remove

CNVs from the result set. Presumably the intent here is that it increases sensitivity at a given

specificity?

Further, this strategy seems likely to produce false negatives as even slightly imperfect normalisation

can easily result in a CNV dosage estimate fractionally above 0.5, especially in the case of single-bin

CNVs that would have only a point estimate of the dosage. Did the authors examine the strategy of using

a statistical model for this element rather than a fixed threshold / cutoff?

> CNV tool comparisons

The manuscript states that a range of parameter settings were used for each tool and the best chosen

for each one.

The parameters used for analysis with each tool are insufficiently described (either here, or where results are

presented elsewhere). However I could not find either the ranges (or values) evaluated or the optimal values

chosen for most of the callers (including SavvyCNV). For example, the bin size is stated as a user selectable

parameter of SavvyCNV but the value chosen is not documented in the results, materials and methods or the

supplementary information. These details are important to ensure reproducibility of the work. Similarly for

the transition probability and number of SVD components - if these are left at their defaults then this could

simply be stated.

A particular problem seems to arise in the selection of bin size. I evaluated SavvyCNV on internal data sets

that we have used to validate CNV detection for exome sequencing. It was apparent in this process that choosing

different bin sizes could make deletions in our truth set appear and disappear from the results. This effect

primarily seems to be driven by the luck of how much of a bin (or how many bins) end up overlapping the event

of interest. If bins are too large, small events cannot be captured, while if they are too small, they become

vulnerable to noise and weak power.

# Software Suggestions

- The output file for SavvyCNV does not contain any headers. This makes it difficult to parse and

process in downstream tools (eg: Pandas, Excel, etc).

- The output file is referred to in the documentation as CSV format but is actually written as a tab

separated file

- The output file associates each call to the binned data file that the call came from. This required

me to add separate tracking outside of SavvyCNV to relate the files to each sample. Although it might

be normal to do this implicitly, eg: by naming the input files by sample, I feel it would be more

friendly if there was a way to identify the actual sample.

**Have the authors made all data and (if applicable) computational code underlying the findings in their manuscript fully available?**

Reviewer #1: **No: **Code and exact calls to tools used in the benchmark have not been made available. As specified in the data availability statement, also the in house dataset used in the benchmark could not be made available due to patient privacy issues.

Reviewer #2: **No: **The authors have explained that part of their validation data set is clinical data that cannot be made public for patient confidentiality reasons. I am comfortable with this personally but obviously journal policy should be applied according to your own discretion.

PLOS authors have the option to publish the peer review history of their article (what does this mean?). If published, this will include your full peer review and any attached files.

Reviewer #1: **Yes: **Matthias Schlesner

Reviewer #2: **Yes: **Simon Sadedin
---

## [Decision Letter · Decision Letter 1]

19 Feb 2022

Dear Dr Laver,

We are pleased to inform you that your manuscript 'SavvyCNV: genome-wide CNV calling from off-target reads' has been provisionally accepted for publication in PLOS Computational Biology.

Best regards,

Mihaela Pertea

Software Editor

PLOS Computational Biology

Mihaela Pertea

Software Editor

PLOS Computational Biology

Reviewer's Responses to Questions

**Comments to the Authors:**

Reviewer #1: The authors have adressed all issues raised by the reviewers.

I have no further comments.

**Have the authors made all data and (if applicable) computational code underlying the findings in their manuscript fully available?**

Reviewer #1: None

PLOS authors have the option to publish the peer review history of their article (what does this mean?). If published, this will include your full peer review and any attached files.

Reviewer #1: **Yes: **Matthias Schlesner

---

## [Editor Report · Acceptance letter]

14 Mar 2022

PCOMPBIOL-D-21-01118R1 

SavvyCNV: genome-wide CNV calling from off-target reads

Dear Dr Laver,

I am pleased to inform you that your manuscript has been formally accepted for publication in PLOS Computational Biology. Your manuscript is now with our production department and you will be notified of the publication date in due course.

With kind regards,

Katalin Szabo
